# How Dietary Factors Affect DNA Methylation: Lesson from Epidemiological Studies

**DOI:** 10.3390/medicina56080374

**Published:** 2020-07-25

**Authors:** Andrea Maugeri, Martina Barchitta

**Affiliations:** Department of Medical and Surgical Sciences and Advanced Technologies “GF Ingrassia”, University of Catania, 95123 Catania, Italy; martina.barchitta@unict.it

**Keywords:** diet, nutritional epidemiology, epigenetics, folate, dietary patterns

## Abstract

Over the past decades, DNA methylation has been proposed as a molecular mechanism underlying the positive or negative effects of diet on human health. Despite the number of studies on this topic is rapidly increasing, the relationship between dietary factors, changes in DNA methylation and health outcomes remains unclear. In this review, we summarize the literature from observational studies (cross-sectional, retrospective, or prospective) which examined the association of dietary factors (nutrients, foods, and dietary patterns) with DNA methylation markers among diseased or healthy people during the lifetime. Next, we discuss the methodological pitfalls by examining strengths and limitations of published studies. Finally, we close with a discussion on future challenges of this field of research, raising the need for large-size prospective studies evaluating the association between diet and DNA methylation in health and diseases for appropriate public health strategies.

## 1. Introduction

‘Let food be thy medicine and medicine be thy food’—since ancient times this quote by Hippocrates has inspired humans to understand how foods could affect our health. However, in the last decades alone, research has revealed that dietary habits play a crucial role in maintaining health and in disease prevention [1]. More recently—especially through the Human Genome Project—it has become clearer how the interaction between genes and diet might influence human health, both positively and negatively. In this scenario, nutritional epigenomics elucidates what mechanisms are involved in the gene–diet interaction by investigating the effects of nutrients, foods and dietary patterns on DNA methylation, histone modifications, and non-coding RNAs [2]. As first defined by Conrad H. Waddington and later updated, these epigenetic mechanisms lead to heritable changes in gene expression that occur without modifications in DNA sequence [3]. Among these, DNA methylation is one of the most extensively studied and best characterized epigenetic mechanisms. In mammals, it is regulated by the activity of DNA methyltransferases (DNMT1, DNMT3a and DNMT3b) and ten-eleven translocation (TET) proteins [4]. DNA methylation almost exclusively occurs within CpG islands, short sequences that typically contain about 5–10 CpG dinucleotides per 100 bp [5]. Although some CpG islands are located within 60% of gene promoters in humans, the majority of them occur in repetitive sequences scattered throughout the genome [6]. In general, DNA methylation is involved in several key physiological processes (e.g., genomic imprinting, X-chromosome inactivation, regulation of gene expression, maintenance of chromosome integrity through chromatin modulation, DNA stabilization and DNA–protein interactions) [7], while aberrant DNMTs functions are associated with cardiovascular diseases, metabolic disorders, neurodegeneration, and cancers [8,9,10,11,12,13,14,15,16].

However, both environmental exposures and lifestyles can potentially modify DNA methylation, leading to genome reprogramming in exposed individuals and in future generations [17]. For instance, in vitro studies have demonstrated how several classes of nutrients modulated DNA methylation process via different mechanisms [18,19,20,21,22,23,24,25,26,27,28,29,30,31]. However, the relationship between the exposome—the sum of internal and external exposures of an individual in a lifetime—and DNA methylation is too complex to be investigated through in vitro models. For this reason, previous epidemiological studies began to focus on the effect of specific foods and nutrients on global and local DNA methylation. More recently, the research is approaching to the study of more complex dietary patterns to determine how they might affect epigenetic mechanisms in the lifetime. Here, we present current evidence obtained through observational research on the relationship between diet—in its broadest sense—and DNA methylation. Notably, findings summarized in this review come from studies conducted on “healthy” or diseased individuals during all stages of life. The search strategy and selection criteria are shown in Figure 1.

## 2. Diet and DNA Methylation: The First Evidence from Cancer Research

The first evidence on the role of epigenetic mechanisms in human diseases comes from the research on cancer, pointing out how neoplastic cell transformation is characterized by profound modifications in global and local DNA methylation [32]. For this reason, the association between diet and DNA methylation was first investigated in cancer patients (Table 1). In fact, this relationship could at least partially explain the molecular basis underpinning the risky or protective effects of foods and nutrients against cancer.

### 2.1. Colon Cancer

In 2003, van Engeland and colleagues suggested how alcohol consumption and folate intake might differentially affect methylation of genes involved in cancer initiation and development. Specifically, they carried out a cross-sectional study on 122 colorectal cancer patients and evaluated promoter methylation status of six genes (i.e., APC, p14, p16, MLH1, O6-MGMT, and RASSF1A) in tumor biopsies. Interestingly, the proportion of hypermethylated genes was higher in patients with high alcohol consumption or low folate intake [33]. In the following years, these dietary factors were perhaps the most extensively investigated and further studies have made meaningful progresses also because of the increasing knowledge on cancer epigenetics [50]. Staying on colorectal cancer, a case-control study conducted in Spain partially supported the preliminary findings described above. Indeed, Mas and colleagues demonstrated that colorectal cancer patients with hypermethylation of p16 reported lower intake of folate—but also vitamin A, vitamin B1, potassium and iron—than controls. Moreover, the intake of vitamin A was significantly lower in patients with hypermethylation of p14 or MLH1 [38]. In contrast, neither folate intake nor alcohol consumption were associated with MLH1 hypermethylation among colorectal cancer patients recruited in a cross-sectional study in the Netherlands [39]. With respect to alcohol, instead, Nishihara and colleagues found that the consumption of more than 15 g/day was associated with higher odds of tumor characterized by IGF2 hypomethylation [46].

Further advances in cancer epigenetics led to the definition of the CpG island methylator phenotype (CIMP), a characteristic of approximately 30–50% of colorectal cancers, in which numerous CpG islands within tumor suppressor genes are methylated [15]. In line, Slattery and colleagues evaluated whether dietary factors were associated with CIMP status in 1154 patients with colorectal cancer. While alcohol consumption was associated with higher odds of CIMP-low tumor (i.e., less than two methylated genes), the intake of folate, vitamins B6 and B12, and methionine was not associated with CIMP status [37]. However, a potential sex-dependent relationship between folate intake and CIMP status in rectal cancer was supposed by Curtin and colleagues. In fact, they found that folate intake was associated with CIMP-high tumor negatively among women and positively among men [41]. More recently, Mehta and colleagues evaluated the association of dietary patterns—which considered the complex combination and interaction of foods—with CIMP status in a cross-sectional study on 1285 colorectal cancer patients. Interestingly, adherence to a western dietary pattern—characterized by the intake of red and processed meats, high-fat dairy products, refined grains, and desserts—was associated with CIMP-low tumor [48].

Another aspect that characterizes neoplastic cells is the progressive loss of DNA methylation at global level [13]. Accordingly, Ferrari and colleagues measured global DNA methylation in 189 patients with colorectal cancer using an enzyme-linked immunosorbent assay. They also evaluated alcohol consumption and the intake of energy, macronutrients, and micronutrients (i.e., folate, vitamin B2, vitamin B6, vitamin B12, choline, betaine, and methionine). However, none of these dietary factors was associated with global DNA methylation in tumor biopsies or blood samples [49]. According to some studies, global DNA methylation could be determined by measuring the methylation level of long interspersed nucleotide element-1 (LINE-1). Although controversies exist on its application as a surrogate marker of global DNA methylation, LINE-1 hypomethylation was associated with higher cancer risk, microsatellite instability (MSI) and CIMP-high status [13,15]. Accordingly, Schernhammer and colleagues investigated the relationship of alcohol consumption and the intake of intake of folate and B vitamins with LINE-1 methylation in patients with colon cancer. In particular, they observed that LINE-1 hypomethylation was more common among patients with lower folate intake and higher alcohol consumption [40].

### 2.2. Gastric Cancer

In 2004, Nan and colleagues carried out a case-control study on 110 patients with gastric cancer and 220 age- and sex-matched controls. Among cancer patients, they evaluated the relationship of food consumption and nutrient intakes with methylation of MLH1 in tumor biopsies. Interestingly, MLH1 hypermethylation was more common among alcohol drinkers and subjects with high consumption of vegetables or low consumption of potato [34]. However, the following year, Yuasa and colleagues failed in demonstrating the association of dietary factors with MLH1 methylation in 73 patients with gastric cancer. In contrast, they demonstrated that the consumption of green tea and cruciferous vegetables negatively correlated with methylation of CDX2 [35]. More recently, Zhang and colleagues investigated the effect of food consumption and nutrient intakes on RUNX3 methylation of tumor biopsies from 184 gastric cancer patients. Firstly, they demonstrated a positive association between egg consumption and RUNX3 methylation. A similar relationship was obtained for nut consumption, but only in patients with *Helicobacter pylori* infection. In contrast, the consumption of fruits, as well as the intake of carbohydrate, vitamin B1, and vitamin E was negatively associated with RUNX3 methylation [44].

### 2.3. Breast Cancer

In 2011, two separate research groups independently investigated the relationship of one-carbon-related micronutrients and compounds with DNA methylation in tumor biopsies of women with breast cancer [42,43]. Tao and colleagues found that the intake of folate, vitamins B2, B6, B12, and methionine was not associated with methylation of E-cadherin, p16, and RAR-β [42]. In contrast, Xu and colleagues demonstrated that dietary factors (i.e., folate, methionine, choline, betaine, B2, B6, B12, and alcohol) were associated with methylation status of a panel of 13 breast cancer genes (i.e., ESR1, PGR, BRCA1, APC, p16, CDH1, RAR-β, TWIST1, CCND2, GSTP1, RASSF1A, HIN1, and DAPK) [43]. In general, the intake of micronutrients was associated with DNA methylation both positively and negatively. Among the 13 genes examined, CCND2, HIN1 and CHD1 were the most sensitive, since their methylation status was associated with intake of at least two dietary factors evaluated. Interestingly, the intake of vitamins B2 and B6 was associated with methylation status of three genes evaluated (CDH1, TWIST1, and HIN1 for vitamin B2, CDH1, CCND2, and HIN1 for vitamin B6) [43]. In line, the study by Pirouzpanah and colleagues confirmed that the intake of folate and vitamin B12 was negatively associated with RAR-β and BRCA1 methylation, while the intake of riboflavin and pyridoxine was positively associated with RAR-β methylation [47].

### 2.4. Other Cancers

In 2006, for the first time, Kraunz and colleagues hypothesized that folate intake was related to p16 methylation among patients with head and neck squamous cell carcinoma. Interestingly, they demonstrated that low intake of folate increased the odds of p16 hypermethylation in tumor biopsies [36]. More recently, Chen and colleagues designed a case-control study on 90 patients with esophageal squamous cell carcinoma and 60 healthy controls to evaluate the relationship between the consumption of roasted meat with p16 methylation of esophageal mucosa tissue. Interestingly, their findings pointed out that daily intake of roasted meat was associated with p16 hypermethylation among cancer patients but not among healthy controls [45]. Finally, we selected the cross-sectional study by Piyathilake and colleagues examining the relationship of dietary patterns with LINE-1 methylation in blood samples from women with abnormal cervical cytology. The authors showed that adherence to the unhealthy dietary pattern was positively associated with the risk of high-grade cervical intraepithelial neoplasia (CIN2+). In contrast, women who adhered to the healthy dietary pattern were more likely to have higher LINE-1 methylation level than those who adhered to the unhealthy diet [19].

## 3. The Effect of Diet on DNA Methylation in “Healthy” People

Beyond the efforts summarized above, other research groups focused their attention on the relationship between diet and DNA methylation before cancer or other diseases developed (Table 2). In 2010, Stidley and colleagues conducted a cross-sectional analysis on 1101 smokers from the Lovelace Smokers Cohort to evaluate the association of dietary factors with DNA methylation of eight genes (i.e., p16, MGMT, DAPK, RASSF1A, PAX5 α, PAX5 β, GATA4 and GATA5) in sputum samples. Interestingly, they noted that high intake of green vegetables and folate was associated with lower DNA methylation, expressed as less than two genes methylated [51]. A prospective analysis on the same cohort further demonstrated both positive (saturated fat) and negative (vitamin A, folate, and vitamin D) associations of dietary factors with DNA methylation of twelve cancer genes [52].

Several studies in this field of research evaluated the effect of dietary factors—and especially folate intake—on DNA methylation in blood samples. The epigenome-wide association study by Mandaviya and colleagues found 74 differentially methylated regions (DMRs) associated with folate intake, and, among them, the most significant was within the LGALS3BP gene [53]. Instead, evidence on global DNA methylation was controversial and characterized by positive, negative or no associations with folate intake. For instance, Ono and colleagues found a negative relationship between folate intake and global DNA methylation assessed by the luminometric methylation assay [20]. Other studies, instead, used LINE-1 and/or Alu sequences as surrogate markers for estimating global DNA methylation level. While two studies failed in demonstrating any effect of nutrients in one-carbon metabolism on LINE-1 or Alu methylation [54,55], others found a positive association. Interestingly, Zhang and colleagues demonstrated that folate intake from fortified foods was positively associated with LINE-1 methylation after adjusting for age, gender, race, BMI, diet, and physical activity [21]. In line, Agodi and colleagues showed that women with folate deficiency, as well as those with low fruit consumption, were more likely to report LINE-1 hypomethylation [22]. The same research groups further demonstrated that adherence to specific dietary patterns might affect LINE-1 methylation in blood samples. In 2011, Zhang and colleagues suggested for the first time that adherence to a healthy dietary pattern was associated with a lower prevalence of LINE-1 hypomethylation [25]. Consistently, Barchitta and colleagues first found that adherence to the Mediterranean diet was positively associated with LINE-1 methylation level [56], and then that this level increased with increasing adherence to a healthy dietary pattern [57]. As suggested by the authors, this relationship could be explained the consumption of healthy foods such as whole-meal cereals, fish, legumes, fruit and vegetables [57]. This was partially in line with findings from the epigenome-wide association study by Nicodemus-Johnson and colleagues. Interestingly, they found more than 5000 CpG sites associated with the consumption of fruit and juice, respectively. Specifically, fruit-specific epigenetic signature could regulate genes associated with antigen presentation and chromosome or telomere maintenance, while the juice-specific epigenetic signatures were related to inflammatory pathways [58].

Only a few studies evaluated how energy consumption and intake of other nutrients affected DNA methylation globally. In particular, Marques-Rocha and colleagues showed that people with high intake of calories, iron and riboflavin, and those with low intake of copper, niacin and thiamin reported higher LINE-1 methylation than their counterparts. However, the authors failed in demonstrating any association of dietary factors with TNF-α and IL-6 methylation [59].

Finally, we selected the study by Shimazu and colleagues, evaluating the effect of fruits, green and yellow vegetables, and salt on DNA methylation of gastric mucosa of 281 subjects with no history of cancer and treatment against *Helicobacter pylori* infection. Specifically, the authors demonstrated that the intake of green and yellow vegetables was associated with lower methylation of miR-124a-3, but not with EMX1 and NKX6-1 genes [60]. 

## 4. Prospects for Studying Obesity, Metabolic Disorders, and Cardiovascular Diseases

More recently, the study of the relationship between diet and DNA methylation focused on the effect on obesity traits and metabolic disorders (Table 3). The methylation status of TNF-α and its association with dietary factors were evaluated among 40 normal weight women in Spain. Hermsdorff and colleagues found that women with high truncal fat had lower TNF-α methylation than those with low truncal adiposity. Moreover, they demonstrated that the intake of n-6 fatty acid was negatively associated with TNF-α methylation [61]. Instead, Carraro and colleagues showed that the intake of fruits and the adherence to the healthy eating index were negatively associated with TNF-α methylation level [62]. The authors proposed that the DNA methylation changes in TNF-α might represent a molecular mechanism underpinnings benefits of healthy dietary pattern and fruit consumption on glucose tolerance [62]. More recently, a cross-sectional study by Ramos-Lopez and colleagues evaluated the relationship of folate intake with genomic methylation profile among 47 obese participants of the Metabolic Syndrome Reduction in Navarra-Spain trial. They found 51 CpGs associated with folate intake, one of which located in the CAMKK2 gene [63]. In general, this gene is implicated in the regulation of metabolic processes (e.g., adiposity and glucose homeostasis) and its methylation level was positively associated with folate intake [63]. The same research group evaluated the associations of dopamine gene methylation patterns, obesity markers, metabolic profiles, and dietary intake in 473 adults from the Methyl Epigenome Network Association project. The authors first found 12 CpG sites that were strongly associated with BMI and abdominal obesity. Among these CpG sites, those that were within the SLC18A1 and SLC6A3 genes correlated with total energy consumption and carbohydrate intake [64]. Lack of evidence exists regarding the relationship between diet and DNA methylation in patients with cardiovascular diseases. To the best of our knowledge, the only examining this topic was the case-control study by Gomez-Uriz and colleagues. The authors evaluated nutrient intake, quality of diet, and methylation of TNF-α and PON among twelve patients with a first episode of parenchymal ischemic stroke and 12 patients with non-vascular neurological disorder [65]. Interestingly, TNF-α methylation was associated with lipid intake and diet quality among non-stroke patients, while PON methylation was associated with energy intake in both groups [65].

## 5. The Relationship between Dietary Factors and DNA Methylation in Mothers and Their Children

Another field of application of studying the interaction between dietary factors and DNA methylation regards the effect of maternal diet on pregnancy outcomes and newborns’ health. To our knowledge, the first evidence of this relationships comes from a study on people who were prenatally exposed to famine during the Dutch Hunger Winter in the middle of the 20th century. Indeed, sixty years later, study participants exhibited lower DNA methylation of the IGF2 gene when compared with unexposed individuals [66]. Further investigations on the same cohort revealed additional DNA methylation changes in genes implicated in metabolic disorders, such as INSIGF2, GNASAS1, MEG3, IL-10 and LEP [67]. More recently, findings from a genome-scale analysis confirmed that prenatal famine exposure was significantly associated with DNA methylation signatures in pathways related to growth and metabolism [68].

Accordingly, several observational studies evaluated the effect of dietary factors (i.e., nutrients, foods, and dietary patterns) on DNA methylation using data and sample from mother–child pairs (Table 4). In 2013, Boeke and colleagues failed in demonstrating an association of maternal intake of methyl donor nutrients with maternal and cord blood LINE-1 methylation [69]. Dietary cadmium, instead, was positively associated with maternal LINE-1 methylation at the first trimester of pregnancy, and negatively with cord blood methylation at birth [69]. Consistently, the study by Taylor and colleagues did not find a significant effect of one-carbon metabolism nutrients on global DNA methylation in cord blood or buccal cells of children [70]. In contrast, Haggarty and colleagues demonstrated that folate intake and use of folic acid supplements were associated with low LINE-1 methylation in the offspring. The authors also observed that folate intake positively associated with IGF2 and negatively with PEG3 methylation [71]. With respect to IGF2, Rijlaarsdam and colleagues found that maternal diet high in fat and carbohydrates before pregnancy was positively associated with IGF2 methylation in offspring [72]. Pauwels and colleagues evaluated the effect of maternal dietary factors before and during pregnancy on DNA methylation of RXRA, LEP, DNMT1, and IGF2. Interestingly, intake of betaine and methionine before pregnancy was positively associated with DNMT and LEP methylation; methyl group donor intake in the second trimester was negatively associated with LEP and DNMT methylation; intake of choline and folate in the third trimester was positively associated with DNMT methylation, and negatively with RXRA methylation [73]. To the best of our knowledge, only the study by McCullough and colleagues investigated the effect of complex dietary patterns rather than specific foods or nutrients. However, the authors demonstrated that a pro-inflammatory diets increased cytokine levels, but no effect on DNA methylation of nine genes was evident (i.e., IGF2, H19, MEG3, MEG3-IG, PEG3, MEST, SGCE/PEG10, NNAT, PLAGL1) [74].

## 6. Methodological Pitfalls and Future Challenges

An evaluation of studies included in the present review revealed some similarities but also great differences in study designs and methods. We noted that most studies had a cross-sectional design (65%), while only 13.9% and 20.9% were case-control or prospective studies, respectively. This did not allow us to understand the causal relationship between dietary factors and changes in DNA methylation, especially for studies on diseased patients. For this reason, further large-scale prospective research is encouraged to uncover the complex relationship of diet with DNA methylation among healthy people, but also to understand whether changes in DNA methylation could represent a molecular mechanism underlying the protective or risky effect of dietary factors against cancer and other diseases.

With respect to dietary assessment, almost all studies (90.7%) collected data using Food Frequency Questionnaire (FFQ), which currently represents the gold standard in nutritional epidemiology. Although this method does not preclude errors and inaccuracies of its measurements, other tools for dietary assessment (e.g., weighted records and 24-h recalls) are also prone to misreporting [75]. In fact, the development of novel dietary assessment tools could help to overcome limitations of traditional assessments [76], however, their costs and unresolved intrinsic problems related to self-reporting prevent their extensive use in epidemiological research. Moreover, only a few studies (25.6%) evaluated the complex mixture of foods into a dietary pattern rather than the consumption of specific foods or nutrients. Indeed, nutritional research is generally moving towards the study of dietary patterns using both a priori (e.g., predefined dietary indexes) or a posteriori (i.e., dimensionality reduction techniques) approaches [77,78,79,80,81,82,83,84,85,86].

Differences between studies also regarded methods used for DNA methylation analysis. Indeed, most studies applied methylation-specific PCR and pyrosequencing of bisulfite-treated DNA (37.2% and 34.9%, respectively), while 16.3% used alternative methods (e.g., enzyme-linked immunosorbent assay, Methylight assay, and mass spectrometry methods). The choice between these techniques followed the strides forward in DNA methylation profiling, but also depended on studies’ objectives, sample size and associated costs. Only in recent years, instead, some studies (11.6%) started applying DNA methylation microarrays based on the Illumina Beadchip technology to analyze more than 450,000 CpG sites. Each of the abovementioned techniques presented strengths and weaknesses that were systematically evaluated by Laird [87]. Nowadays, however, advances in DNA methylation microarrays along with reduction in their costs make them a versatile approach to explore the association between dietary factors and DNA methylation status, also in large-size studies [88].

Another issue to be considered when interpreting findings of this kind of study regards the significance of DNA methylation changes observed. While for many genes and genomic sequences it is clear their function and probable association with health and diseases, for others—especially for DMRs identified by epigenome-wide association studies—it is necessary to understand what they exactly entail. Indeed, the biological effect of a particular DNA methylation change is often unknown and unpredictable based on our current knowledge. In support of this, functional studies using in vivo models might help to investigate the significance of DNA methylation changes and to solve controversies on the effect of specific nutrients and foods. For instance, examining studies included in the current review, it was not clear if folate intake was positively or negatively associated with DNA methylation. In fact, it is likely that folates—as well as other methyl donors—are important for DNA methylation maintenance, while other nutrients and/or bioactive foods exert their influence directly on enzymes involved in the methylation process. In fact, several reviews properly summarized evidence of this influence coming from in vivo studies [89,90]. Thus, in addition to investigate the association between dietary factors and DNA methylation, a deeper insight into the consequences of epigenetic changes on physiological and pathological events should be recommended. Studies on LINE-1 methylation represented a prime example of this need. Indeed, while methylation status of LINE-1 sequences could be considered a surrogate marker of global DNA methylation [13], the impacts of both hypo-and hyper-methylation should be considered. Indeed, aberrant methylation of LINE-1 sequences was previously associated with aging, cancer, neurodegeneration, metabolic disorders, and obesity by affecting genome and chromosome stability [13,14,16,24,91,92,93,94,95]. Moreover, since DNA methylation is highly cell-and tissue-specific, differences in tissue samples or blood cell composition should be addressed when comparing results from different studies. Last but not least, DNA methylation is highly sensitive to environmental exposure-in its broadest and most comprehensive meaning–and genetic factors [17,20,56,96,97,98,99,100,101,102]. For this reason, future studies should comprehensively evaluate not only dietary factors and their relationship with DNA methylation, but also the effects of other lifestyles, social determinants, environmental exposures, and genetic variants that might affect the methylation process. Finally, investments are necessary to promote studies able to assess the potential applications of genome knowledge into clinical practice to improve public health [103].

## 7. Discussion

In this review, we summarize several epidemiological studies that report the relationship between dietary factors and DNA methylation, providing compelling insight into the possible effects of our diet in health and diseases. In general, DNA methylation—the most common covalent modification of the human genome—is related to transcriptional repression through chromatin-remodeling complexes. Accordingly, it is crucial for the control of gene expression during the development, as well as for imprinting, X-inactivation, cancer and other diseases [104]. Specifically, the aging process is also accompanied by altered DNA methylation, which in turn has an effect on health-span [105].

Remarkable advances in epigenetics in the past decades—especially in cancer epigenetics—have coincided with increasing understanding of the impact of diet and lifestyles on certain diseases, such as obesity, metabolic disorders, cancer, and neurogenerative diseases [106]. However, the molecular mechanisms underpinning these effects are not yet well understood. The first evidence of the epigenome–diet interaction comes from cancer research and several studies proposed how the protective or risky impact of diet could be modulated by changes in DNA methylation profiles. However, the causal link between dietary factors, altered DNA methylation and cancer development remains unclear at present, raising questions to guide future research. Several cross-sectional analyses on healthy people suggested that nutrients and foods, as well as their combination into more complex dietary patterns, might be associated with global and specific DNA methylation. The next step would be a better comprehension of the effects of DNA methylation changes on health outcomes. Thus, there is the need for large-size prospective studies evaluating the association between diet and DNA methylation before cancer and/or other disease develop, and which subsequently assess the risk associated with DNA methylation changes.

A similar approach is exploited by the research on the transgenerational impact of maternal diet on newborns’ health, also considering food safety [107]. Indeed, findings from birth cohorts proposed several epigenetic mechanisms as biological explanations for early-life exposures on later health [108]. Despite studies on this topic are continuously increasing, many doubts remain with regard to the strength of the evidence to date.

Although the research has made great strides, our review suggests that there are significant challenges to be faced. Many of them relate to methodological features to collect, analyze and interpret data. Notably, the putative mechanism by which DNA methylation affects health outcomes is modulating RNA expression. For this reason, future studies should correlate DNA methylation changes, mRNA expression in the relevant genes, and their biological effects. While it is relatively straightforward to recognize DNA methylation changes with the available techniques, it is more difficult to understand how these changes associate with functional significance. However, it is worth mentioning that DNA methylation changes are not the only molecular mechanisms associated with health and diseases. Indeed, both genetic variants and other epigenetic signatures (i.e., histone modifications and altered miRNA expression) might be related to dietary factors and in turn affect human health [109,110,111,112,113,114]. Moreover, it is becoming important to simultaneously assess different social factors, environmental exposures, and behaviors that dependently or independently modulate the risk for cancers, metabolic disorders, cardiovascular and neurodegenerative diseases throughout life [81,115,116,117,118,119,120,121,122]. Based on these considerations, there is a need for large-size prospective studies evaluating the association between diet and DNA methylation in the context of a wider range of molecular changes and human exposome. This applies to research conducted on healthy or diseased people.

## 8. Conclusions

In conclusion, our review discusses the effects of dietary factors on DNA methylation changes and the increasingly knowledge of how these changes might affect health and diseases. However, it also raises concerns due to methodological pitfalls that limit the strength of current evidence. Although DNA methylation will provide an explanation to understand molecular mechanisms underlying the effect of diet on our health in the future, more efforts will be necessary to decipher this relationship.

## Figures and Tables

**Figure 1 medicina-56-00374-f001:**
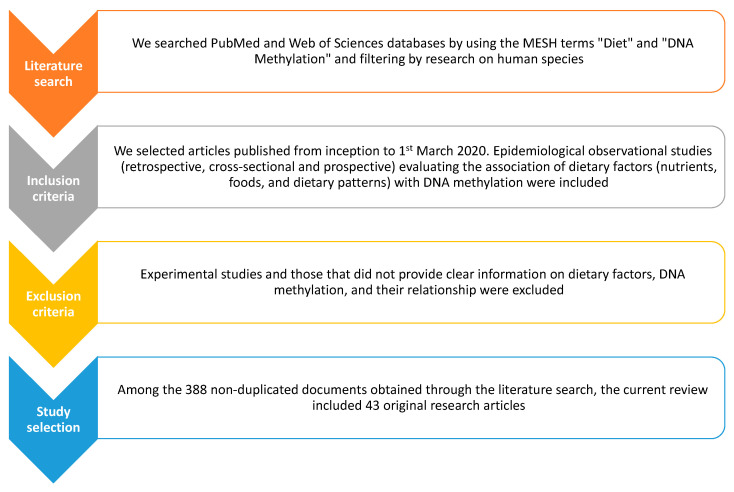
Search strategy and selection criteria of epidemiological studies examining the association between dietary factors and DNA methylation.

**Table 1 medicina-56-00374-t001:** Summary of studies examining the relationship between dietary factors and DNA methylation in cancer patients.

First Author and Year of Publication	Country	Study Design	Study Population	Dietary Factors	DNA Methylation Markers	Sample Type	DNA Methylation Method	Main Findings
van Engeland et al. 2003 [33]	The Netherlands	Cross-sectional	122 patients with colorectal cancer	Alcohol and folate	APC, p14, p16, MLH1, MGMT, and RASSF1A	Tumor biopsy	Methylation Specific PCR	For each gene, prevalence of promoter hypermethylation was higher in patients with low folate intake and high alcohol consumption. The number of patients with at least one gene methylated was higher in the low folate intake/high alcohol intake group than their counterparts
Nan et al. 2004 [34]	Korea	Case Control	110 patients with gastric cancer and 220 age- and sex-matched controls	Foods, calories, nutrients, vitamins, and minerals	MLH1	Tumor biopsy	Methylation Specific PCR	Alcohol consumption was associated with higher odds of MLH1 hypermethylation. High consumption of vegetables and low consumption of potato were associated with higher odds of MLH1 hypermethylation
Yuasa et al. 2005 [35]	Japan	Cross-sectional	73 patients with gastric cancer	Foods and nutrients	CDX2, p16 and MLH1	Tumor biopsy	Methylation Specific PCR	Among men, consumption of green tea and cruciferous vegetables was negatively correlated with CDX2 methylation
Kraunz et al. 2006 [36]	USA	Cross-sectional	242 patients with head and neck squamous cell carcinoma	Folate	p16	Tumor biopsy	Methylation Specific PCR	Low intake of folate was associated with higher odds of p16 methylation
Slattery et al. 2006 [37]	USA	Case Control	1154 patients with colon cancer and 1256 controls	Cruciferous vegetables, alcohol, folate, vitamins B6 and B12, methionine, and fiber	CIMP	Tumor biopsy	Methylation Specific PCR	Alcohol consumption was associated with higher odds of CIMP-low. The intake of fiber was associated with CIMP status
Mas et al. 2007 [38]	Spain	Case Control	120 patients with colorectal cancer and 296 controls	Nutrients	p16, p14 and MLH1	Tumor biopsy	Methylation Specific PCR	Patients with low intake of folate, vitamin A, vitamin B1, potassium and iron showed lower p16 methylation than controls. Patients with low vitamin A intake showed lower p14 and MLH1 methylation
De Vogel et al. 2008 [39]	The Netherlands	Cross-sectional	648 patients with colorectal cancer	Folate, vitamin B2 and vitamin B6, methionine and alcohol	MLH1	Tumor biopsy	Methylation Specific PCR	Intakes of folate, vitamin B2, methionine and alcohol were not associated with MLH1 hypermethylation. Among men, intake of vitamin B6 was associated with MLH1 hypermethylation
Schernhammer et al. 2009 [40]	USA	Cross-sectional	609 patients with colon cancer	Alcohol, folate and B vitamins	LINE-1	Tumor biopsy	Pyrosequencing	Participants with higher folate intake were less likely to exhibit LINE-1 hypomethylation. Alcohol consumption was positively associated with LINE-1 hypomethylation
Curtin et al. 2011 [41]	USA	Case Control	951 patients with rectal cancer and 1205 controls	Folate, riboflavin, vitamins B6 andB12, and methionine	CIMP	Tumor biopsy	Methylation Specific PCR	Women with higher folate intake had lower odds of CIMP+ phenotype. Men with higher folate intake had higher odds of CIMP+ tumor
Tao et al. 2011 [42]	USA	Cross-sectional	1170 women with breast cancer	One-carbon-related micronutrients and compounds	E-cadherin, p16, and RAR-β	Tumor biopsy	Methylation Specific PCR	Dietary intake of folate, vitamins B2, B6, B12, and methionine was not associated with methylation of E- cadherin, p16, and RAR-β
Xu et al. 2011 [43]	USA	Cross-sectional	851 women with breast cancer	One-carbon-related micronutrients and compounds	13 breast cancer-related genes	Tumor biopsy	Methylation Specific PCR and Methyight assay	Intake of B2 and B6 correlated with promoter methylation status in 3 out of the 13 breast cancer genes evaluated. Both positive (hypermethylation) and inverse (hypomethylation) associations were observed
Piyathilake et al. 2012 [19]	USA	Cross-sectional	319 women with abnormal cervical cytology	Dietary patterns	LINE-1	Blood	Pyrosequencing	Women with healthy dietary pattern were more likely to have higher LINE-1 methylation than those who adhered to an unhealthy dietary pattern
Zhang et al. 2013 [44]	South Korea	Cross-sectional	184 patients with gastric cancer	Calories, foods, nutrients, vitamins and minerals	RUNX3	Tumor biopsy	Methylation Specific PCR	High consumption of eggs was associated with higher odds of RUNX3 methylation. High consumption of nuts was associated with higher odds of RUNX3 methylation in patients with *Helicobacter pylori* infection. High consumption of fruits and high intake of carbohydrate, vitamin B1, and vitamin E was associated with lower odds of RUNX3 methylation
Chen et al. 2014 [45]	China	Case Control	90 patients with esophageal squamous cell carcinoma and 60 healthy adults	Roast meat	p16	Esophageal mucosa tissue	Pyrosequencing	Consumption of roast meat was positively associated with p16 methylation among cases. No association was evident among healthy subjects
Nishihara et al. 2014 [46]	USA	Cross-sectional	993 patients with colorectal cancer from the Nurses’ Health Study and the Health Professionals’ Follow-up Study	Alcohol, vitamin B6, vitamin B12, folate, and methionine	IGF2	Tumor biopsy	Pyrosequencing	Consumption of >15 g alcohol/d was associated with higher risk of colorectal cancer with lower IGF2 methylation levels. The association of vitamin B-6, vitamin B-12, and folate intakes with cancer risk did not significantly differ according to IGF2 methylation level
Pirouzpanah et al. 2015 [47]	Iran	Cross-sectional	149 women with breast cancer	Folate, vitamins B2, B6, B12, and methionine	RAR-β, BRCA1 and RASSF1A	Tumor biopsy	Methylation Specific PCR	Intake of folate and vitamin B12 was negatively associated with RAR-β and BRCA1 methylation. Intake of riboflavin and pyridoxine was positively associated with RAR-β methylation
Mehta et al. 2017 [48]	USA	Cross-sectional	1285 patients with colorectal cancer from the Health Professionals’ Follow-up Study and the Nurses’ Health Study	Dietary patterns	CIMP	Tumor biopsy	Pyrosequencing	Adherence to the western dietary pattern, characterized by red and processed meats, high-fat dairy products, refined grains, and desserts, was associated with CIMP-low phenotype
Ferrari et al. 2019 [49]	Brazil	Cross-sectional	189 patients with colon or rectal adenocarcinoma	Alcohol, folate, vitamins B2, B6, and B12, choline, betaine, methionine, energy, carbohydrate, protein, and lipid	Global DNA methylation	Tumor biopsy and blood	Enzyme-linked immunosorbent assay	No association between dietary intakes and global DNA methylation was evident

**Table 2 medicina-56-00374-t002:** Summary of studies examining the relationship between dietary factors and DNA methylation in healthy people.

First Author and Year of Publication	Country	Study Design	Study Population	Dietary Factors	DNA Methylation Markers	Sample Type	DNA Methylation Method	Main Findings
Stidley et al. 2010 [51]	USA	Cross-sectional	1101 smokers from the Lovelace Smokers Cohort	Total and animal fat, vitamin C, vitamin E, folate, carotene, alpha carotene, beta-carotene, lycopene, lutein and zeaxanthin, and retinol	p16, MGMT, DAPK, RASSF1A, PAX5 α, PAX5 β, GATA4 and GATA5	Sputum	Methylation Specific PCR	High intake of folate was associated with lower DNA methylation
Zhang et al. 2011 [54]	USA	Cross-sectional	161 cancer-free individuals	Folate, vitamins B12 and B6, riboflavin and methionine	LINE-1	Blood	Pyrosequencing	No association between intake of nutrients in one-carbon metabolism and LINE-1 methylation
Ono et al. 2012 [20]	Japan	Cross-sectional	384 healthy women	Folate and vitamins B2, B6, and B12	Global DNA methylation	Blood	Methylight assay	Folate intake was negatively associated with global DNA methylation
Zhang et al. 2012 [21]	USA	Cross-sectional	180 cancer-free individuals	Folate and dietary patterns	IL-6 and LINE-1	Blood	Pyrosequencing	Folate intake was positively associated with LINE-1 methylation
Perng et al. 2014 [55]	USA	Cross-sectional	1002 participants of the Multi-Ethnic Study of Atherosclerosis study	Folate, vitamins B12 and B6, zinc, and methionine	LINE-1 and Alu	Blood	Pyrosequencing	Intake of methyl-donor micronutrients was not associated with DNA methylation
Agodi et al. 2015 [22]	Italy	Cross-sectional	177 healthy women	Mediterranean Diet and folate	LINE-1	Blood	Pyrosequencing	Women with low consumption of fruit and those with folate deficiency were more likely to exhibit LINE-1 hypomethylation
Shimazu et al. 2015 [60]	Japan	Cross-sectional	281 subjects without cancer and with no history of treatment against *Helicobacter pylori* infection	Green/yellow vegetables, fruit and salt	miR-124a-3, EMX1 and NKX6-1	Gastric mucosa	Methylation Specific PCR	Intake of green/yellow vegetables was negatively associated with methylation of miR-124a-3
Marques-Rocha et al. 2016 [59]	Brazil	Cross-sectional	156 subjects without metabolic disease, chronic inflammation, hydric balance disorders, changes in body composition and problems in nutrient absorption or metabolism	Energy and nutrients	LINE-1, TNF-α and IL-6	Blood	Methylation-sensitive high-resolution melting analysis	Individuals with higher LINE-1 methylation had higher daily intakes of calories, iron and riboflavin, and lower intakes of copper, niacin and thiamin
Nicodemus-Johnson et al. 2017 [58]	USA	Prospective	2148 Caucasian individuals from the Framingham Heart Study Offspring cohort	Fruits and juices	Genomic methylation profile	Blood	Infinium Illumina Human Methylation 450 k BeadChip arrays	There were 5221 and 5434 CpG sites associated with the intake of fruit and juice, respectively
Barchitta et al. 2018 [56]	Italy	Cross-sectional	299 healthy women	Mediterranean diet	LINE-1	Blood	Pyrosequencing	Adherence to the Mediterranean diet was positively associated with LINE-1 methylation level
Leng et al. 2018 [52]	Mexico	Prospective	327 Hispanics and 1502 non-Hispanic White smokers from the Lovelace Smokers Cohort	Nutrients	12 tumor suppressor genes	Sputum	Methylation Specific PCR	Intake of vitamin A, folate, and vitamin D was negatively associated with DNA methylation levels. Intake of saturated fat was positively associated with DNA methylation levels
Barchitta et al. 2019 [57]	Italy	Cross-sectional	349 healthy women	Foods and dietary patterns	LINE-1	Blood	Pyrosequencing	Consumption of whole-meal bread, cereals, fish, fruit, raw and cooked vegetables, legumes, soup, potatoes, fries, rice and pizza were positively correlated with LINE-1 methylation. LINE-1 methylation level increased with increasing adherence to a prudent dietary pattern
Mandaviya et al. 2019 [53]	Netherlands, Italy, Finland, USA, UK	Prospective	5841 participants with no history of cancer from 10 cohorts	Folate and vitamin B-12	Genomic methylation profile	Blood	Infinium Illumina Human Methylation 450 k BeadChip arrays	74 folate-associated DMRs, of which 73 were negatively associated with folate intake. The most significant folate-associated DMR was a 400-base pair (bp) spanning region annotated to the LGALS3BP gene
Zhang et al. 2011 [25]	USA	Cross-sectional	149 cancer-free individuals	Dietary patterns	LINE-1	Blood	Pyrosequencing	Adherence to a prudent dietary pattern was associated with a lower prevalence of LINE-1 hypomethylation. No association between the Western dietary pattern and LINE-1 methylation was evident

**Table 3 medicina-56-00374-t003:** Summary of studies examining the relationship between dietary factors and DNA methylation and their association with obesity traits and/or cardiometabolic disorders.

First Author and Year of Publication	Country	Study Design	Study Population	Dietary Factors	DNA Methylation Markers	Sample Type	DNA Methylation Method	Main Findings
Hermsdorff et al. 2013 [61]	Spain	Cross-sectional	40 normal-weight women	Energy and fat	TNF-α	Blood	Epityper Methylation Analysis	Women with high truncal fat showed lower TNF-α methylation than those with lower truncal adiposity. Intake of n-6 fatty acid was negatively associated with TNF-α methylation
Gomez-Uriz et al. 2014 [65]	Spain	Case Control	12 patients with a first episode of parenchymal ischemic stroke and 12 patients with non-vascular neurological disorder	Nutrients and indexes of quality of diet	TNF-α and PON	Blood	Matrix-Assisted Laser Desorption/Ionization-Time Of Flight (MALDI-TOF) mass spectrometry	TNF-α methylation was related to lipid intake and dietary indexes in non-stroke patients. PON methylation was related to energy intake and quality of the diet
Carraro et al. 2016 [62]	Spain	Cross-sectional	40 normal-weight healthy women	Fruit and Healthy Eating Index	TNF-α	Blood	MALDI-TOF mass spectrometry	Healthy eating index was negatively associated with TNF-α methylation level. A higher intake of fruits was associated with lower TNF-α methylation
Ramos-Lopez et al. 2017 [63]	Spain	Cross-sectional	47 obese adults from the Metabolic Syndrome Reduction in Navarra, Spain trial	Folate	Genomic methylation profile	Blood	Infinium Illumina Human Methylation 450 k BeadChip arrays	A total of 51 CpGs were associated with folate intake, including one located in the CAMKK2 gene. Folate deficiency was related to lower CAMKK2 methylation
Ramos-Lopez et al. 2019 [64]	Spain	Cross-sectional	247 adults from the Methyl Epigenome Network Association project	Energy, carbohydrates, protein, and fat	Genomic methylation profile	Blood	Infinium Illumina Human Methylation 450 k BeadChip arrays	Methylation of SLC18A1, SLC6A3, and SLC6A3 correlated with total energy consumption and carbohydrate intake

**Table 4 medicina-56-00374-t004:** Summary of studies examining the relationship between dietary factors and DNA methylation in pregnant women and their children.

First Author and Year of Publication	Country	Study Design	Study Population	Dietary Factors	DNA Methylation Markers	Sample Type	DNA Methylation Method	Main Findings
Boeke et al. 2012 [69]	USA	Prospective	830 mother–child pairs	Vitamin B12, betaine, choline, folate, cadmium, zinc and iron	LINE-1	Maternal and infant cord blood	Pyrosequencing	No association of maternal intake of methyl donor nutrients with maternal and cord blood methylation. Periconceptional betaine intake was inversely associated with cord blood methylation; dietary cadmium was positively associated with first trimester methylation and inversely with cord blood methylation
Haggarty et al. 2013 [71]	United Kingdom	Prospective	913 mother–child pairs	Folate intake	PEG3, IGF2, small nuclear ribonucleoprotein polypeptide N, and LINE-1	Infant cord blood	Pyrosequencing	Folate intake was positively associated with IGF2 methylation and negatively with PEG3 and LINE-1 methylation in the offspring
McCullough et al. 2017 [74]	USA	Prospective	338 mother–child pairs from the NEST cohort	Dietary inflammatory potential	IGF2, H19, MEG3, MEG3-IG, PEG3, MEST, SGCE/PEG10, NNAT, PLAGL1	Infant cord blood	Pyrosequencing	Pro-inflammatory diets increased cytokine levels, but no association between dietary inflammatory potential and DNA methylation was evident
Pauwels et al. 2017 [73]	Belgium	Prospective	115 mother–child pairs from the Maternal Nutrition and Offspring’s Epigenome study	Betaine, choline, folate, and methionine	Global DNA methylation and RXRA, LEP, DNMT1, and IGF2	Infant cord blood	Liquid chromatography–tandem mass spectrometry and Pyrosequencing	Before pregnancy, intakes of betaine and methionine were positively associated with DNMT and LEP methylation. In the second trimester, methyl group donor intake was negatively associated with LEP and DNMT methylation. In the last trimester, intake of choline and folate was positively associated with DNMT methylation and negatively with RXRA methylation
Rijlaarsdam et al. 2017 [72]	UK	Prospective	346 mother–child pairs from the Avon Longitudinal Study of Parents and Children	Dietary patterns	IGF2	Infant cord blood and blood at 7 years	Infinium Illumina Human Methylation 450 k BeadChip arrays	Maternal diet high in fat and carbohydrates before pregnancy was positively associated with IGF2 methylation at birth
Taylor et al. 2017 [70]	Australia	Prospective	73 children from the WATCH study	Methionine, folate, vitamins B2, B6 and B12 and choline	Global DNA methylation	Buccal cells	Enzyme-linked immunosorbent assay	No association between one-carbon metabolism nutrient intake and global DNA methylation levels was evident

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
