# Peer review of "How Dietary Factors Affect DNA Methylation: Lesson from Epidemiological Studies"

_medicina, 2020, doi:10.3390/medicina56080374_

Round 1
Reviewer 1 Report
In their manuscript entitled “How dietary factors affect DNA methylation: lesson from epidemiological studies”, Maugeri and Barchitta discuss the current evidence on dietary factors to affect cellular epigenome.
This is a well-written and timely manuscript that summarizes the knowledge from a number of clinical trials as well as epidemiological evidence from both cancer patients and “healthy” subjects.
This reviewer would like to provide some recommendations for the article:
- Very often, there is no compelling evidence from the reviewed and discussed studies; i.e., those on folates/folic acid as the results are somewhat controversial. In these regards, it might be helpful to refer to animal studies as they me be helpful to draw more meaningful conclusions.
- It would be helpful to provide information on methodologies that were utilized to asses DNA methylation: it is an open secret that the result often depend on which method of DNA methylation was used as well as bioinformatics analysis. A section that would discuss these methodologies and their advantages/disadvantages would be very helpful.
- While the authors mention in the abstract their vision for large-size prospective studies to evaluate the association between diet and DNA methylation, the Discussion section clearly misses this component. Adding the authors’ perspectives on this matter would is highly encouraged.
Author Response
Dear Editor,
this document is intended for the convenience of the editor and reviewers and contains the list of the requested changes. We would like to take this opportunity to thank the reviewers for the insightful comments and suggestions to improve the paper. We hereby submit to your attention a revised version of the manuscript in which we have considered all comments. The following list of changes and answers to comments of Reviewers addresses all changes made in the manuscript (in red font).
Comments submitted by Reviewer 1
In their manuscript entitled “How dietary factors affect DNA methylation: lesson from epidemiological studies”, Maugeri and Barchitta discuss the current evidence on dietary factors to affect cellular epigenome.
This is a well-written and timely manuscript that summarizes the knowledge from a number of clinical trials as well as epidemiological evidence from both cancer patients and “healthy” subjects.
Answer: We are very grateful with Reviewer 1 for his/her helpful comments and suggestions which helped us in improving our manuscript.
Very often, there is no compelling evidence from the reviewed and discussed studies; i.e., those on folates/folic acid as the results are somewhat controversial. In these regards, it might be helpful to refer to animal studies as they me be helpful to draw more meaningful conclusions.
Answer: As reported by Reviewer 1, controversies exist on the effect of nutrients and foods on DNA methylation. Accordingly, we discussed on this issue in lines 315-322, also citing previous reviews which summarized evidence from animal studies.
It would be helpful to provide information on methodologies that were utilized to asses DNA methylation: it is an open secret that the result often depend on which method of DNA methylation was used as well as bioinformatics analysis. A section that would discuss these methodologies and their advantages/disadvantages would be very helpful.
Answer: We agree with this suggestion and hence we included information on methods used for DNA methylation analysis in all the tables. We also discussed on differences between techniques in lines 298-309, also citing relevant studies.
While the authors mention in the abstract their vision for large-size prospective studies to evaluate the association between diet and DNA methylation, the Discussion section clearly misses this component. Adding the authors’ perspectives on this matter would is highly encouraged.
Answer: As suggested, we added our perspectives and the need for large-size studies in lines 373-382.
Reviewer 2 Report
The review is focused, densely argued and appropriately referenced, combining both recent references and the citation of earlier work. It is generally well written and quite enlightening; the tables are clear and informative. Maugeri et al. have nicely portrayed the role of nutritional factors and its link to DNA methylation and associated diseases. I enjoyed reviewing it. I have no major comments, just minor suggestions:
- Paragraph 5 should describe the studies conducted on one of the most elegant models of periconceptional calorie restriction, represented by the famine exposure during the Dutch Hunger Winter. Heijmans and colleagues [https://doi.org/10.1073/pnas.0806560105] demonstrated that the insulin-like growth factor 2 (IGF2) locus was differentially methylated in the white blood cells of individuals prenatally exposed to famine. Further investigations have also revealed altered DNA methylation in other genes in the famine exposed-group (INSIGF2, GNASAS1, MEG3, IL-10 and LEP), some of which have an identified role in metabolic disorders progression [https://doi.org/10.1093/hmg/ddp353]. Finally, they reported a genome-scale analysis of differential DNA methylation in whole blood after periconceptional exposure to famine [https://doi.org/10.1038/ncomms6592].
- Please check the use of semicolons in the abstract section.
- The term 'authors' is written in both uppercase and lowercase. Please make it uniform along the manuscript.
Author Response
Dear Editor,
this document is intended for the convenience of the editor and reviewers and contains the list of the requested changes. We would like to take this opportunity to thank the reviewers for the insightful comments and suggestions to improve the paper. We hereby submit to your attention a revised version of the manuscript in which we have considered all comments. The following list of changes and answers to comments of Reviewers addresses all changes made in the manuscript (in red font).
Comments submitted by Reviewer 2
The review is focused, densely argued and appropriately referenced, combining both recent references and the citation of earlier work. It is generally well written and quite enlightening; the tables are clear and informative. Maugeri et al. have nicely portrayed the role of nutritional factors and its link to DNA methylation and associated diseases. I enjoyed reviewing it. I have no major comments, just minor suggestions:
Answer: We are very grateful with Reviewer 1 for his/her helpful comments and suggestions which helped us in improving our manuscript.
Paragraph 5 should describe the studies conducted on one of the most elegant models of periconceptional calorie restriction, represented by the famine exposure during the Dutch Hunger Winter. Heijmans and colleagues [https://doi.org/10.1073/pnas.0806560105] demonstrated that the insulin-like growth factor 2 (IGF2) locus was differentially methylated in the white blood cells of individuals prenatally exposed to famine. Further investigations have also revealed altered DNA methylation in other genes in the famine exposed-group (INSIGF2, GNASAS1, MEG3, IL-10 and LEP), some of which have an identified role in metabolic disorders progression [https://doi.org/10.1093/hmg/ddp353]. Finally, they reported a genome-scale analysis of differential DNA methylation in whole blood after periconceptional exposure to famine [https://doi.org/10.1038/ncomms6592].
Answer: We agree that studies on individuals exposed to famine during the Dutch Hunger Winter represent elegant model to study the effect of dietary restrictions on DNA methylation. However, suggested studies were excluded since they did not meet inclusion criteria of our review (i.e. these studies did not really evaluate the effect of dietary factors such as nutrients, foods or dietary patterns). Anyway, we cited these studies in the first lines of paragraph 5 (lines 243-251).
Please check the use of semicolons in the abstract section.
Answer: We are sorry for this mistake that has been corrected.
The term 'authors' is written in both uppercase and lowercase. Please make it uniform along the manuscript.
Answer: As suggested, we made consistent the term “Author” throughout the text.